# BEVFusion: A Simple and Robust LiDAR-Camera Fusion Framework

**Tingting Liang**[1*]  **Hongwei Xie**[2*]  **Kaicheng Yu**[2*]  **Zhongyu Xia**[1]  **Zhiwei Lin**[1]
**Yongtao Wang**[1§]  **Tao Tang**[2&3]  **Bing Wang**[2]  **Zhi Tang**[1]

[1] Wangxuan Institute of Computer Technology, Peking University, China
[2] DAMO Academy, Alibaba Group, China
[3] Shenzhen Campus of Sun Yat-sen University, China

{tingtingliang, xiazhongyu, zhiweilin,wyt,tangzhi}@pku.edu.cn
{hongwei.xie.90, kaicheng.yu.yt trent.tangtao, blucewang6}@gmail.com

## Abstract

Fusing the camera and LiDAR information has become a de-facto standard for 3D object detection tasks. Current methods rely on point clouds from the LiDAR sensor as queries to leverage the feature from the image space. However, people discovered that this underlying assumption makes the current fusion framework infeasible to produce any prediction when there is a LiDAR malfunction, regardless of minor or major. This fundamentally limits the deployment capability to realistic autonomous driving scenarios. In contrast, we propose a surprisingly simple yet novel fusion framework, dubbed BEVFusion, whose camera stream does not depend on the input of LiDAR data, thus addressing the downside of previous methods. We empirically show that our framework surpasses the state-of-the-art methods under the normal training settings. Under the robustness training settings that simulate various LiDAR malfunctions, our framework significantly surpasses the state-of-the-art methods by 15.7% to 28.9% mAP. To the best of our knowledge, we are the first to handle realistic LiDAR malfunction and can be deployed to realistic scenarios without any post-processing procedure. The code is available at https://github.com/ADLab-AutoDrive/BEVFusion.

## 1  Introduction

Vision-based perception tasks, like detecting bounding boxes in 3D space, have been a critical aspect of fully autonomous driving tasks [56, 42, 57, 41]. Among all the sensors of a traditional vision on-vehicle perception systems, LiDAR and camera are usually the two most critical sensors that provide accurate point cloud and image features of a surrounding world. In the early stage of perception systems, people design separate deep models for each sensor [37, 38, 59, 16, 53], and fuse the information via post-processing approaches [32]. Note that, people discover that bird's eye view (BEV) has been an de-facto standard for autonomous driving scenarios as, generally speaking, car cannot fly [20, 23, 39, 16, 54, 33]. However, it is often difficult to regress 3D bounding boxes on pure image inputs due to the lack of depth information, and similarly, it is difficult to classify objects on point clouds when LiDAR does not receive enough points.

Recently, people have designed LiDAR-camera fusion deep networks to better leverage information from both modalities. Specifically, the majority of works can be summarized as follow: i) given one or a few points of the LiDAR point cloud, LiDAR to world transformation matrix and the essential matrix (camera to world); ii) people transform the LiDAR points [43, 46, 47, 46, 17, 60] or proposals

---

§Corresponding Author.
*Equal Contribution.

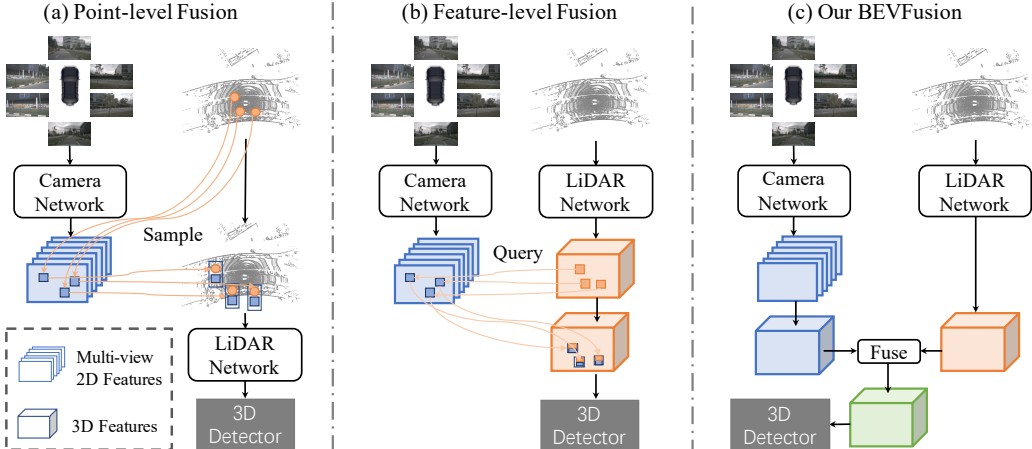

Figure 1: **Comparison of our framework with previous LiDAR-camera fusion methods.** Previous fusion methods can be broadly categorized into (a) point-level fusion mechanism [43, 46, 47, 46, 17, 60] that project image features onto raw point clouds, and (b) feature-level fusion mechanism [4, 61, 1, 21] that projects LiDAR feature or proposals on each view image separately to extract RGB information. (c) In contrast, we propose a novel yet surprisingly simple framework that disentangles the camera network from LiDAR inputs.

[4, 61, 1, 21] into camera world and use them as queries, to select corresponding image features. This line of work constitutes the state-of-the-art methods of 3D BEV perception.

However, one underlying assumption that people overlooked is, that as one needs to generate image queries from LiDAR points, the current LiDAR-camera fusion methods intrinsically depend on the raw point cloud of the LiDAR sensor, as shown in Fig. 1. In the realistic world, people discover that if the LiDAR sensor input is missing, for example, LiDAR points reflection rate is low due to object texture, a system glitch of internal data transfer, or even the field of view of the LiDAR sensor cannot reach 360 degrees due to hardware limitations [62], current fusion methods fail to produce meaningful results[1]. This fundamentally hinders the applicability of this line of work in the realistic autonomous driving system.

We argue the ideal framework for LiDAR-camera fusion should be, that each model for a single modality should not fail regardless of the existence of the other modality, yet having both modalities will further boost the perception accuracy. To this end, we propose a surprisingly simple yet effective framework that disentangles the LiDAR-camera fusion dependency of the current methods, dubbed BEVFusion. Specifically, as in Fig. 1 (c), our framework has two independent streams that encode the raw inputs from the camera and LiDAR sensors into features within the same BEV space. We then design a simple module to fuse these BEV-level features after these two streams, so that the final feature can be passed into modern task prediction head architecture [20, 59, 1].

As our framework is a general approach, we can incorporate current single modality BEV models for camera and LiDAR into our framework. We moderately adopt Lift-Splat-Shoot [34] as our camera stream, which projects multi-view image features to the 3D ego-car coordinate features to generate the camera BEV feature. Similarly, for the LiDAR stream, we select three popular models, two voxel-based ones and a pillar-based one [59, 1, 20] to encode the LiDAR feature into the BEV space.

On the nuScenes dataset, our simple framework shows great generalization ability. Following the same training settings [20, 59, 1], BEVFusion improves PointPillars and CenterPoint by 18.4% and 7.1% in mean average precision (mAP) respectively, and achieves a superior performance of 69.2% mAP comparing to 68.9% mAP of TransFusion [1], which is considered as state-of-the-art. Under the robust setting by randomly dropping the LiDAR points inside object bounding boxes with a probability of 0.5, we propose a novel augmentation technique and show that our framework surpasses

---

[1]See [62] and Sec. 4.4 for more details.

all baselines significantly by a margin of 15.7% ∼28.9% mAP and demonstrate the robustness of our approach.

Our contribution can be summarized as follow: i) we identify an overlooked limitation of current LiDAR-camera fusion methods, which is the dependency of LiDAR inputs; ii) we propose a simple yet novel framework to disentangle LiDAR camera modality into two independent streams that can generalize to multiple modern architectures; iii) we surpass the state-of-the-art fusion methods under both normal and robust settings.

## 2 Related Works

Here, we categorize the 3D detection methods broadly based on their input modality.

**Camera-only.** In the autonomous driving domain, detecting 3D objects with only camera-input has been heavily investigated in recent years thanks to the KITTI benchmark [11]. Since there is only one front camera in KITTI, most of the methods have been developed to address monocular 3D detection [29, 40, 28, 19, 63, 66, 39, 49, 48]. With the development of autonomous driving datasets that have more sensors, like nuScenes[2] and Waymo [44], there exists a trend of developing methods [50, 51, 53] that take multi-view images as input and found to be significantly superior to monocular methods. However, voxel processing is often accompanied by high computation.

As in common autonomous driving datasets, Lift-Splat-Shoot (LSS) [34] uses depth estimation network to extract the implied depth information of multi-perspective images and transform camera feature maps into 3D Ego-car coordinate. Methods [39, 16, 54] are also inspired by LSS [34] and refer to the LiDAR for the supervision on depth prediction. A similar idea can also be found in BEVDet [16, 15], the state-of-the-art method in multi-view 3D object detection. MonoDistill [7] and LiGA Stereo [12] improve performance by unify LiDAR information to a camera branch.

**LiDAR-only.** LiDAR methods initially lie in two categories based on their feature modality: i) point-based methods that directly operate on the raw LiDAR point clouds [38, 37, 36, 42, 57, 22]; and ii) transforming the original point clouds into a Euclidean feature space, such as 3D voxels [65] and feature pillar [20, 52, 59]. Recently, people started to exploit these two feature modalities in a single model to increase the representation power [4, 64, 41]. Another line of work is to exploit the benefit of the bird's eye view plane [20, 10, 45].

**LiDAR-camera fusion.** As the features produced by LiDAR and camera contain complementary information in general, people started to develop methods that can be jointly optimized on both modalities and soon become a de-facto standard in 3D detection. As in Fig. 1, these methods can be divided into two categories depending on their fusion mechanism, (a) point-level fusion where one queries the image features via the raw LiDAR points and then concatenates them back as additional point features [17, 43, 47, 60]; (b) feature-level fusion where one firstly projects the LiDAR points into a feature space [61] or generates proposals [1], queries the associated camera features then concatenates back to the feature space [24, 6]. The latter constitutes the state-of-the-art methods in 3D detection, specifically, TransFusion [1] uses the bounding box prediction of LiDAR features as a proposal to query the image feature, then adapts a Transformer-like architecture to fuse the information back to LiDAR features. Similarly, DeepFusion [21] projects LiDAR features on each view image as queries and then leverages cross-attention for two modalities.

An overlooked assumption of the current fusion mechanism is they heavily rely on the LiDAR point clouds, in fact, if the LiDAR input is missing, these methods will inevitably fail. This will hinder the deployment of such algorithms in realistic settings. In contrast, our BEVFusion is a surprisingly simple yet effective fusion framework that fundamentally overcomes this issue by disentangling the camera branch from the LiDAR point clouds as shown in Fig. 1(c). In addition, concurrent works [27, 58] also address this problem and propose effective LiDAR-camera 3D perception models.

**Other modalities.** There exist other works to leverage other modalities, such as fusing camera-radar by feature map concatenation [3, 18, 31, 30]. While interesting, these methods are beyond the scope of our work. Despite a concurrent work [5] aims to fuse multi-modalities information in a single network, its design is limited to one specific detection head [53] while our framework can be generalized to arbitrary architectures.

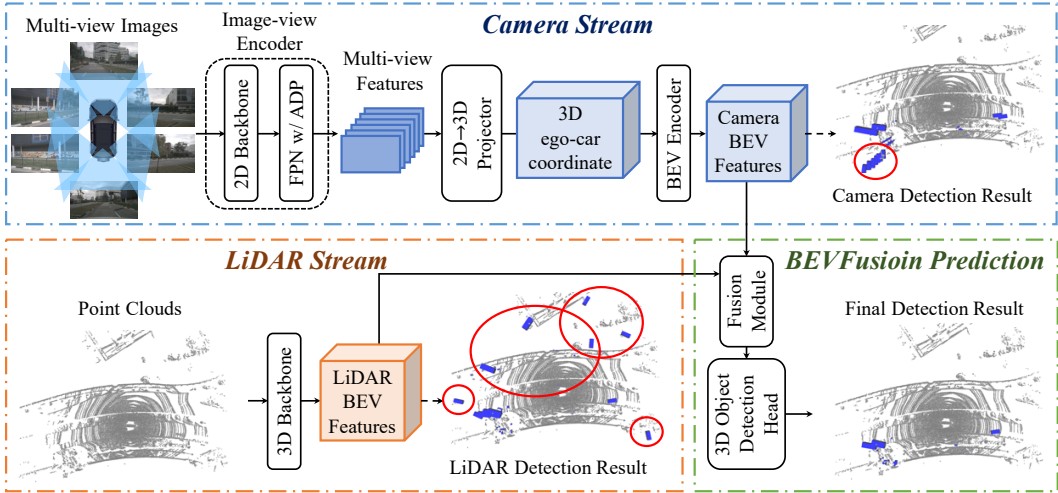

Figure 2: **An overview of BEVFusion framework.** With point clouds and multi-view image inputs, two streams separately extract features and transform them into the same BEV space: i) the camera-view features are projected to the 3D ego-car coordinate features to generate camera BEV feature; ii) 3D backbone extracts LiDAR BEV features from point clouds. Then, a fusion module integrates the BEV features from two modalities. Finally, a task-specific head is built upon the fused BEV feature and predicts the target values of 3D objects. In detection result figures, blue boxes are predicted bounding boxes, while red circled ones are the false positive predictions.

## 3 BEVFusion: A General Framework for LiDAR-Camera Fusion

As shown in Fig. 2, we present our proposed framework, BEVFusion, for the 3D object detection in detail. As our fundamental contribution is disentangling the camera network from the LiDAR features, we first introduce the detailed architecture of the camera and LiDAR stream, then present a dynamic fusion module to incorporate features from these modalities.

### 3.1 Camera stream architecture: From multi-view images to BEV space

As our framework has the capability to incorporate any camera streams, we begin with a popular approach, Lift-Splat-Shoot (LSS) [35]. As the LSS is originally proposed for BEV semantic segmentation instead of 3D detection, we find out that directly using the LSS architecture has inferior performance, hence we moderately adapt the LSS to improve the performance (see Sec. 4.5 for ablation study). In Fig. 2 (top), we detail the design of our camera stream in the aspect of an image-view encoder that encodes raw images into deep features, a view projector module that transforms these features into 3D ego-car coordinate, and an encoder that finally encodes the features into the bird's eye view (BEV) space.

**Image-view Encoder** aims to encode the input images into semantic information-rich deep features. It consists of a 2D backbone for basic feature extraction and a neck module for scale variate object representation. Different from LSS [34] which uses the convolutional neural network ResNet [13] as the backbone network, we use the more representative one, CB-Swin-Tiny [25] as the backbone. Following [34], we use a standard Feature Pyramid Network (FPN) [26] on top of the backbone to exploit the features from multi-scale resolutions. To better align these features, we first propose a simple feature *Adaptive Module* (ADP) to refine the upsampled features. Specifically, we apply an adaptive average pooling and a $1 \times 1$ convolution for each upsampled feature before concatenating. See Appendix Sec. A for the detailed module architecture.

**View Projector Module.** As the image features are still in 2D image coordinate, we design a view projector module to transform them into 3D ego-car coordinate. We apply $2D \rightarrow 3D$ view projection proposed in [34] to construct the Camera BEV feature. The adopted view projector takes the image-view feature as input and densely predicts the depth through a classification manner. Then, according

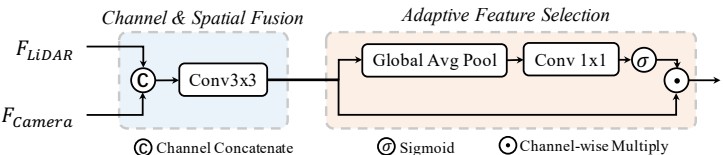

Figure 3: **Dynamic Fusion Module.**

to camera extrinsic parameters and the predicted image depth, we can derive the image-view features to render in the predefined point cloud and obtain a pseudo voxel $V \in R^{X \times Y \times Z \times C}$.

**BEV Encoder Module.** To further encode the voxel feature $V \in R^{X \times Y \times Z \times C}$ into the BEV space feature ($\mathbf{F}_{\text{Camera}} \in R^{X \times Y \times C_{\text{Camera}}}$), we design a simple encoder module. Instead of applying pooling operation or stacking 3D convolutions with stride 2 to compress $z$ dimension, we adopt the *Spatial to Channel (S2C) operation* [54] to transform $V$ from 4D tensor to 3D tensor $V \in R^{X \times Y \times (ZC)}$ via reshaping to preserve semantic information and reduce cost. We then use four $3 \times 3$ convolution layers to gradually reduce the channel dimension into $C_{\text{Camera}}$ and extract high-level semantic information. Different from LSS [34] which extracts high-level features based on downsampled low-resolution features, our encoder directly processes full-resolution Camera BEV features to preserve the spatial information.

### 3.2 LiDAR stream architecture: From point clouds to BEV space

Similarly, our framework can incorporate any network that transforms LiDAR points into BEV features, $\mathbf{F}_{\text{LiDAR}} \in R^{X \times Y \times C_{\text{LiDAR}}}$, as our LiDAR streams. A common approach is to learn a parameterized voxelization [65] of the raw points to reduce the Z-dimension and then leverage networks consisting of sparse 3D convolution [56] to efficiently produce the feature in the BEV space. In practice, we adopt three popular methods, PointPillars [20], CenterPoint [59] and TransFusion [1] as our LiDAR stream to showcase the generalization ability of our framework.

### 3.3 Dynamic fusion module

To effectively fuse the BEV features from both camera ($\mathbf{F}_{\text{Camera}} \in R^{X \times Y \times C_{\text{Camera}}}$) and LiDAR ($\mathbf{F}_{\text{LiDAR}} \in R^{X \times Y \times C_{\text{LiDAR}}}$) sensors, we propose a dynamic fusion module in Fig. 3. Given two features under the same space dimension, an intuitive idea is to concatenate them and fuse them with learnable static weights. Inspired by Squeeze-and-Excitation mechanism [14], we apply a simple channel attention module to select important fused features. Our fusion module can be formulated as:

$$\mathbf{F}_{\text{fused}} = f_{\text{adaptive}}(f_{\text{static}}([\mathbf{F}_{\text{Camera}}, \mathbf{F}_{\text{LiDAR}}])), \tag{1}$$

where $[\cdot, \cdot]$ denotes the concatenation operation along the channel dimension. $f_{\text{static}}$ is a static channel and spatial fusion function implemented by a $3 \times 3$ convolution layer to reduce the channel dimension of concatenated feature into $C_{\text{LiDAR}}$. With input feature $\mathbf{F} \in R^{X \times Y \times C_{\text{LiDAR}}}$, $f_{\text{adaptive}}$ is formulated as:

$$f_{\text{adaptive}}(\mathbf{F}) = \sigma\left(\mathbf{W} f_{\text{avg}}(\mathbf{F})\right) \cdot \mathbf{F}, \tag{2}$$

where $\mathbf{W}$ denotes linear transform matrix (e.g., 1x1 convolution), $f_{\text{avg}}$ denotes the global average pooling and $\sigma$ denotes sigmoid function.

### 3.4 Detection head

As the final feature of our framework is in BEV space, we can leverage the popular detection head modules from earlier works. This is further evidence of the generalization ability of our framework. In essence, we compare our framework on top of three popular detection head categories, anchor-based [20], anchor-free-based [59], and transform-based [1].

## 4 Experiments

In this section, we present our experimental settings and the performance of BEVFusion to demonstrate the effectiveness, strong generalization ability, and robustness of the proposed framework.

Table 1: **Generalization ability of BEVFusion.** We validate the effectiveness of our fusion framework on nuScenes validation set, compared to single modality streams over three popular methods [20, 59, 1]. Note that each method here defines the structure of the LiDAR stream and associated detection head while the camera stream remains the same as in Sec. 3.1.

| Modality | | PointPillars | | CenterPoint | | TransFusion-L | |
|---|---|---|---|---|---|---|---|
| Camera | LiDAR | mAP | NDS | mAP | NDS | mAP | NDS |
| ✓ | | 22.9 | 31.1 | 27.1 | 32.1 | 22.7 | 26.1 |
| | ✓ | 35.1 | 49.8 | 57.1 | 65.4 | 64.9 | 69.9 |
| ✓ | ✓ | 53.5 | 60.4 | 64.2 | 68.0 | 67.9 | 71.0 |

## 4.1 Experimental settings

**Dataset.** We conduct comprehensive experiments on a large-scale autonomous-driving dataset for 3D detection, nuScenes [2]. Each frame contains six cameras with surrounding views and one point cloud from LiDAR. There are up to 1.4 million annotated 3D bounding boxes for 10 classes. We use nuScenes detection score (NDS) and mean average precision (mAP) as evaluation metrics. See Appendix B.1 for more details.

**Implementation details.** We implement our network in PyTorch using the open-sourced MMDetection3D [8]. We conduct BEVFusion with CB-Swin-Tiny [25] as 2D backbone for image-view encoder. PointPillars [20], CenterPoint [59], and TransFusion-L [1] are chosen as our LiDAR stream and 3D detection head. We set the image size to $448 \times 800$ and the voxel size following the official settings of the LiDAR stream [20, 59, 1]. Our training consists of two stages: i) We first train the LiDAR stream and camera stream with multi-view image input and LiDAR point clouds input, respectively. Specifically, we train both streams following their LiDAR official settings in MMDetection3D [8]; ii) We then train BEVFusion for another 9 epochs that inherit weights from two trained streams. Note that no data augmentation (i.e., flipping, rotation, or CBGS [67]) is applied when multi-view image input is involved. In this version of paper, we additionally train the fusion phase with BEV-space data augmentation following [16, 27] when comparing with the state-of-the-art methods. In particular, we add the BEV-space augmentation implemented by [27] [2] for a better result. During testing, we follow the settings of LiDAR-only detectors [20, 59, 1] in MMDetection3D [8] without any extra post-processing. See Appendix Sec. B.1 for the detailed hyper-parameters and settings.

## 4.2 Generalization ability

To demonstrate the generalization ability of our framework, we adapt three popular LiDAR-only detector as our LiDAR stream and detection head, PointPillars [20], CenterPoint [59] and TransFusion-L [1], as described in Sec. 3. If not specified, all experimental settings follow their original papers. In Table 1, we present the results of training two single modality streams, followed by jointly optimized. Empirical results show that our BEVFusion framework can significantly boost the performance of these LiDAR-only methods. Despite the limited performance of the camera stream, our fusion scheme improves PointPillars by 18.4% mAP and 10.6% NDS, and CenterPoint and TransFusion-L by a margin of 3.0% ~ 7.1% mAP. This evidences that our framework can generalize to multiple LiDAR backbone networks.

As our method relies on a two-stage training scheme, we nonetheless report the performance of a single stream in the bottom part of Table 1. We observe that the LiDAR stream constantly surpasses the camera stream by a significant margin. We ascribe this to the LiDAR point clouds providing robust local features about object boundaries and surface normal directions, which are essential for accurate bounding box prediction.

## 4.3 Comparing with the state-of-the-art methods

Here, we use TransFusion-L as our LiDAR stream and present the results on the test set of nuScenes in Table 2. Without any test time augmentation or model ensemble, our BEVFusion surpasses all previous LiDAR-camera fusion methods and achieves the state-of-the-art performance of 69.2% mAP

---

[2]https://github.com/mit-han-lab/bevfusion

Table 2: **Results on the nuScenes validation (top) and test (bottom) set.**

| Method | Modality | mAP | NDS | Car | Truck | C.V. | Bus | Trailer | Barrier | Motor. | Bike | Ped. | T.C. |
|---|---|---|---|---|---|---|---|---|---|---|---|---|---|
| FUTR3D [5] | LC | 64.2 | 68.0 | 86.3 | 61.5 | 26.0 | 71.9 | 42.1 | 64.4 | 73.6 | 63.3 | 82.6 | 70.1 |
| BEVFusion | LC | 67.9 | 71.0 | 88.6 | 65.0 | 28.1 | 75.4 | 41.4 | 72.2 | 76.7 | 65.8 | 88.7 | 76.9 |
| BEVFusion* | LC | 69.6 | 72.1 | 89.1 | 66.7 | 30.9 | 77.7 | 42.6 | 73.5 | 79.0 | 67.5 | 89.4 | 79.3 |
| PointPillars[20] | L | 30.5 | 45.3 | 68.4 | 23.0 | 4.1 | 28.2 | 23.4 | 38.9 | 27.4 | 1.1 | 59.7 | 30.8 |
| CBGS[67] | L | 52.8 | 63.3 | 81.1 | 48.5 | 10.5 | 54.9 | 42.9 | 65.7 | 51.5 | 22.3 | 80.1 | 70.9 |
| CenterPoint[59]† | L | 60.3 | 67.3 | 85.2 | 53.5 | 20.0 | 63.6 | 56.0 | 71.1 | 59.5 | 30.7 | 84.6 | 78.4 |
| TransFusion-L [1] | L | 65.5 | 70.2 | 86.2 | 56.7 | 28.2 | 66.3 | 58.8 | 78.2 | 68.3 | 44.2 | 86.1 | 82.0 |
| PointPainting[46] | LC | 46.4 | 58.1 | 77.9 | 35.8 | 15.8 | 36.2 | 37.3 | 60.2 | 41.5 | 24.1 | 73.3 | 62.4 |
| 3D-CVF[61] | LC | 52.7 | 62.3 | 83.0 | 45.0 | 15.9 | 48.8 | 49.6 | 65.9 | 51.2 | 30.4 | 74.2 | 62.9 |
| PointAugmenting[47]† | LC | 66.8 | 71.0 | 87.5 | 57.3 | 28.0 | 65.2 | 60.7 | 72.6 | 74.3 | 50.9 | 87.9 | 83.6 |
| MVP[60] | LC | 66.4 | 70.5 | 86.8 | 58.5 | 26.1 | 67.4 | 57.3 | 74.8 | 70.0 | 49.3 | 89.1 | 85.0 |
| FusionPainting[55] | LC | 68.1 | 71.6 | 87.1 | 60.8 | 30.0 | 68.5 | 61.7 | 71.8 | 74.7 | 53.5 | 88.3 | 85.0 |
| TransFusion[1] | LC | 68.9 | 71.7 | 87.1 | 60.0 | 33.1 | 68.3 | 60.8 | 78.1 | 73.6 | 52.9 | 88.4 | **86.7** |
| BEVFusion (Ours) | LC | **69.2** | **71.8** | **88.1** | 60.9 | **34.4** | **69.3** | **62.1** | **78.2** | 72.2 | 52.2 | **89.2** | 85.2 |
| BEVFusion (Ours)* | LC | **71.3** | **73.3** | **88.5** | **63.1** | **38.1** | **72.0** | **64.7** | **78.3** | **75.2** | **56.5** | **90.0** | **86.5** |

† These methods exploit double-flip during the test time. The best and second best results are marked in **red** and **blue**.

Notion of class: Construction vehicle (C.V.), pedestrian (Ped.), traffic cone (T.C.). Notion of modality: Camera (C), LiDAR (L).

\* These methods exploit BEV-space data augmentation during training.

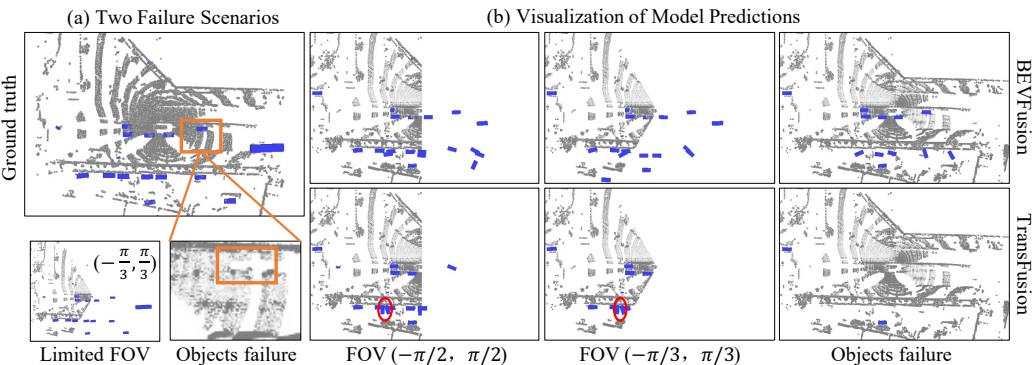

Figure 4: **Visualization of predictions under robustness setting.** (a) We visualize the point clouds under the BEV perspective of two settings, limited field-of-view (FOV) and LiDAR fails to receive object reflection points, where the orange box indicates the object points are dropped. Blue boxes are bounding boxes and red-circled boxes are false-positive predictions. (b) We show the predictions of the state-of-the-art method, TransFusion, and ours under three settings. Obviously, the current fusion approaches fail inevitably when the LiDAR input is missing, while our framework can leverage the camera stream to recover these objects.

comparing to the 68.9% mAP of TransFusion[1]. Note that we do not conduct data augmentation when multi-view image input is involved, while data augmentation plays a critical part in other cutting-edge methods. It is worth noticing that the original TransFusion[1] is a two-stage detector, whose model consists of two independent detection heads. By contrast, our BEVFusion with TransFusion-L as LiDAR backbone only contains one detection head, yet still outperforms the two-stage baseline by 0.3% mAP. As the only difference between our framework and TransFusion is the fusion mechanism, we ascribe this performance gain to a comprehensive exploration of the multi-modality modeling power of BEVFusion.

## 4.4 Robustness experiments

Here, we demonstrate the robustness of our method against all previous baseline methods on two settings, LiDAR and camera malfunctioning. See [62] for more details.

Table 3: **Results on robustness setting of limited LiDAR field-of-view.** Our method significantly boosts the performance of LiDAR-only methods over all settings. Note that, compared to the TransFusion with camera fusion, our method still achieves over 15.3% mAP and 6.6% NDS improvement, showcasing the robustness of our approach.

| FOV | Metrics | PointPillars | | CenterPoint | | TransFusion | | |
|---|---|---|---|---|---|---|---|---|
| | | LiDAR | ↑BEVFusion | LiDAR | ↑BEVFusion | LiDAR | ↑BEVFusion | LC |
| $(-\pi/2,$ | mAP | 12.4 | 36.8 (+24.4) | 23.6 | 45.5 (+21.9) | 27.8 | 46.4 (+18.6) | 31.1 |
| $\pi/2)$ | NDS | 37.1 | 45.8 (+8.7) | 48.0 | 54.9 (+6.9) | 50.5 | 55.8 (+5.3) | 49.2 |
| $(-\pi/3,$ | mAP | 8.4 | 33.5 (+25.1) | 15.9 | 40.9 (+25.0) | 19.0 | 41.5 (+22.5) | 21.0 |
| $\pi/3)$ | NDS | 34.3 | 42.1 (+7.8) | 43.5 | 49.9 (+6.4) | 45.3 | 50.8 (+5.5) | 41.2 |

Table 4: **Results on robustness setting of object failure cases.** Here, we report the results of baseline and our method that trained on the nuScenes dataset with and without the proposed robustness augmentation (Aug.). All settings are the same as in Table 3.

| Aug. | Metrics | Pointpillars | | Centerpoint | | Transfusion | | |
|---|---|---|---|---|---|---|---|---|
| | | LiDAR | ↑BEVFusion | LiDAR | ↑BEVFusion | LiDAR | ↑BEVFusion | LC |
| | mAP | 12.7 | 34.3 (+21.6) | 31.3 | 40.2 (+8.9) | 34.6 | 40.8 (+6.2) | 38.1 |
| | NDS | 36.6 | 49.1 (+12.5) | 50.7 | 54.3 (+3.6) | 53.6 | 56.0 (+2.4) | 55.4 |
| ✓ | mAP | - | 41.6 (+28.9) | - | 54.0 (+22.7) | - | 50.3 (+15.7) | 37.2 |
| ✓ | NDS | - | 51.9 (+15.3) | - | 61.6 (+10.9) | - | 57.6 (+4.0) | 51.1 |

#### 4.4.1 Robustness experiments against LiDAR Malfunctions

To validate the robustness of our framework, we evaluate detectors under two LiDAR malfunctions: i) when the LiDAR sensor is damaged or the LiDAR scan range is restricted, i.e., semi-solid lidars; ii) when objects cannot reflect LiDAR points. We provide visualization of these two failure scenarios in Fig. 4 (a) and evaluate detectors on nuScenes validation set.

**Data augmentation for robustness.** We propose two data augmentation strategies for above two scenarios: i) we simulate the LiDAR sensor failure situation by setting the points with limited Field-of-View (FOV) in range $(-\pi/3, \pi/3)$, $(-\pi/2, \pi/2)$. ii) To simulate the object failure, we use a dropping strategy where each frame has a 0.5 chance of dropping objects, and each object has a 0.5 chance of dropping the LiDAR points inside it. Below, we finetune detectors with these two data augmentation strategies, respectively.

**LiDAR sensor failure.** The nuScenes dataset provides a Field-of-View (FOV) range of $(-\pi, \pi)$ for LiDAR point clouds. To simulate the LiDAR sensor failure situation, we adopt the first aforementioned robust augmentation strategy in Table 3. Obviously, the detector performance degrades as the LiDAR FOV decreases. However, when we fuse camera stream, with the presence of corruptions, the BEVFusion models, in general, are much more robust than their LiDAR-only counterparts, as shown in Fig. 4 (b). Notably, for PointPillars, the mAP increases by 24.4% and 25.1% when LiDAR FOV in $(-\pi/2, \pi/2)$, $(-\pi/3, \pi/3)$, respectively. As for TransFusion-L, BEVFusion improves its LiDAR stream by a large margin of over 18.6% mAP and 5.3% NDS. The vanilla LiDAR-camera fusion approach [1] proposed in TransFusion (denoted as LC in Table 3 and Table 4) heavily relies on LiDAR data, and the gain is limited to less than 3.3% mAP while NDS is decreased. The results reveal that fusing our camera stream during training and inference compensates for the lack of LiDAR sensors to a substantial extent.

**LiDAR fails to receive object reflection points.** Here exist common scenarios when LiDAR fails to receive points from the object. For example, on rainy days, the reflection rate of some common objects is below the threshold of LiDAR hence causing the issue of object failure [62]. To simulate such a scenario, we adopt the second aforementioned robust augmentation strategy on the validation set. As shown in Table 4, when we directly evaluate detectors trained without robustness augmentation, BEVFusion shows higher accuracy than the LiDAR-only stream and vanilla LiDAR-camera fusion approach in TransFusion. When we finetune detectors on the robust augmented training set, BEVFusion largely improves PointPillars, CenterPoint, and TransFusion-L by 28.9%,

Table 5: **Results on robustness setting of camera failure cases.** F denotes front camera.

| Approach | Clean | | Missing F | | Preserve F | | Stuck | |
|---|---|---|---|---|---|---|---|---|
| | mAP | NDS | mAP | NDS | mAP | NDS | mAP | NDS |
| DETR3D[53] | 34.9 | 43.4 | 25.8 | 39.2 | 3.3 | 20.5 | 17.3 | 32.3 |
| PointAugmenting[47] | 46.9 | 55.6 | 42.4 | 53.0 | 31.6 | 46.5 | 42.1 | 52.8 |
| MVX-Net[43] | 61.0 | 66.1 | 47.8 | 59.4 | 17.5 | 41.7 | 48.3 | 58.8 |
| TransFusion[1] | 66.9 | 70.9 | 65.3 | 70.1 | 64.4 | 69.3 | 65.9 | 70.2 |
| BEVFusion | 67.9 | 71.0 | 65.9 | 70.7 | 65.1 | 69.9 | 66.2 | 70.3 |

Table 6: **Ablating the camera stream.**

| BE | ADP | LB | mAP↑ | NDS↑ |
|---|---|---|---|---|
| | | | 13.9 | 24.5 |
| ✓ | | | 17.9 | 27.0 |
| ✓ | ✓ | | 18.0 | 27.1 |
| ✓ | ✓ | ✓ | 22.9 | 31.1 |

BE: our simple BEV Encoder. ADP: adaptive
module in FPN. LB: a large backbone.

Table 7: **Ablating Dynamic Fusion Module.**

| CSF | AFS | Pointpillars | | Centerpoint | | Transfusion | |
|---|---|---|---|---|---|---|---|
| | | mAP↑ | NDS↑ | mAP↑ | NDS↑ | mAP↑ | NDS↑ |
| | | 35.1 | 49.8 | 57.1 | 65.4 | 64.9 | 69.9 |
| ✓ | | 51.6 | 57.4 | 63.0 | 67.4 | 67.3 | 70.5 |
| ✓ | ✓ | 53.5 | 60.4 | 64.2 | 68.0 | 67.9 | 71.0 |

Dynamic Fusion Module consists of CSF and AFS. CSF: channel& spatial
fusion ( Fig. 3(left)). AFS: adaptive feature selection (Fig. 3(right)).

22.7%, and 15.7% mAP. Specifically, the vanilla LiDAR-camera fusion method in TransFusion has a gain of only 2.6% mAP, which is smaller than the performance before finetuning, we hypothesize the reason is that the lack of foreground LiDAR points brings wrong supervision during training on the augmented dataset. The results reveal that fusing our camera stream during training and inference largely compensates for the lack of object LiDAR points to a substantial extent. We provide visualization in Fig. 4 (b).

### 4.4.2 Robustness against camera malfunctions

We further validate the robustness of our framework against camera malfunctions under three scenarios in [62]: i) front camera is missing while others are preserved; ii) all cameras are missing except for the front camera; iii) 50% of the camera frames are stuck. As shown in Table 5, BEVFusion still outperforms camera-only [53] and other LiDAR-camera fusion methods [47, 43, 1] under above scenarios. The results demonstrate the robustness of BEVFusion against camera malfunctions.

### 4.5 Ablation

Here, we ablate our design choice of the camera stream and the dynamic fusion module.

**Components for camera stream.** We conduct ablation experiments to validate the contribution of each component of our camera stream using different components in Table 6. The naive baseline with ResNet50 and feature pyramid network as multi-view image encoder, ResNet18 as BEV encoder following LSS [34], PointPillars [20] as detection head only obtains 13.9% mAP and 24.5% NDS. As shown in Table 6, there are several observations: (i) when we replace the ResNet18 BEV encoder with our simple BEV encoder module, the mAP and NDS are improved by 4.0% and 2.5%. (2) Adding the adaptive feature alignment module in FPN helps improve the detection results by 0.1%. (3) Concerning a larger 2D backbone, i.e., CB-Swin-Tiny, the gains are 4.9% mAP and 4.0% NDS. The camera stream equipped with PointPillars finally achieves 22.9% mAP and 31.1% NDS, showing the effectiveness of our design for the camera stream.

**Dynamic fusion module.** To illustrate the performance of our fusing strategy for two modalities, we conduct ablation experiments on three different 3D detectors, PointPillars, CenterPoint, and TransFusion and the LiDAR stream as baseline. As shown in 7, with a simple channel& spatial fusion (left part in Fig. 3), BEVFusion greately improves its LiDAR stream by 16.5% ($35.1\% \rightarrow 51.6\%$) mAP for PointPillars, 5.9% ($57.1\% \rightarrow 63.0\%$) mAP for CenterPoint, and 2.4% ($64.9\% \rightarrow 67.3\%$) mAP for TransFusion. When adaptive feature selection (right part in Fig. 3) is adopted, the mAP can be futher improved by 1.9%, 1.2%, and 0.6% mAP for PointPillars, CenterPoint, and TransFusion,

respectively. The results demonstrate the necessity of fusing camera and LiDAR BEV features and effectiveness of our Dynamic Fusion Module in selecting important fused features.

## 5   Conclusion

In this paper, we introduce BEVFusion, a surprisingly simple yet unique LiDAR-camera fusion framework that disentangles the LiDAR-camera fusion dependency of previous methods. Our framework comprises two separate streams that encode raw camera and LiDAR sensor inputs into features in the same BEV space, followed by a simple module to fuse these features such that they can be passed into modern task prediction head architectures. The extensive experiments demonstrate the strong robustness and generalization ability of our framework against the various camera and LiDAR malfunctions. We hope that our work will inspire further investigation of robust multi-modality fusion for the autonomous driving task.

## Broader Impacts Statement and Limitations

This paper studies robust LiDAR-camera fusion for 3D object detection. Since the detection explored in this paper is for generic objects and does not pertain to specific human recognition, so we do not see potential privacy-related issues. However, the biased-to-training-data model may pose safety threats when applied in practice. The research may inspire follow-up studies or extensions, with potential applications in autonomous driving tasks. While our study adopts a simple camera stream as the baseline, we also encourage the community to expand the architecture, e.g, with temporal multi-view camera input and alignment between intermediate LiDAR and camera features. We leave the extension of our method towards building such systems for future work.

## Acknowledgment

We thank the anonymous reviewers for their careful reading of our manuscript and their many insightful comments and suggestions.

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
