# Appendix

The supplementary document is organized as follows:

- Sec. A depicts the detailed network architectures of our adaptive module in FPN.
- Sec. B provides the implementation details of BEVFusion.
- Sec. C discusses the effect of Dynamic Fusion Module under robustness settings and reports more robustness analysis on both modality malfunctions, and inferior image conditions.
- Sec. D discusses the performance gain based on the object distance range.
- Sec. E provides the latency and memory footprint of BEVFusion.
- Sec. F provides more visualization results for failure cases and analysis.

## A   Network architectures

The detailed architecture of the proposed adaptive module on top of FPN is shown in Fig. 5. Adaptive modules are applied on top of the standard Feature Pyramid Network (FPN) [26]. For each view image with an input shape of $H \times W \times 3$, the backbone and FPN output multi-scale features $F_2, F_3, F_4, F_5$ with shapes of $H/4 \times W/4 \times C, H/8 \times W/8 \times C, H/16 \times W/16 \times C, H/32 \times W/32 \times C$. The adaptive module upsamples the multi-scale features into shape $H/4 \times W/4 \times C$ via "`torch.nn.Upsample`", "`torch.nn.AdaptiveAvgPool2d`" operations, and a $1 \times 1$ convolution layer. Then, the sampled features are concatenated together followed by a $1 \times 1$ convolution layer to get this view image feature with a shape of $H/4 \times W/4 \times C$. Our model can benefit from the FPN concatenation mechanisms in recent works such as [16, 54].

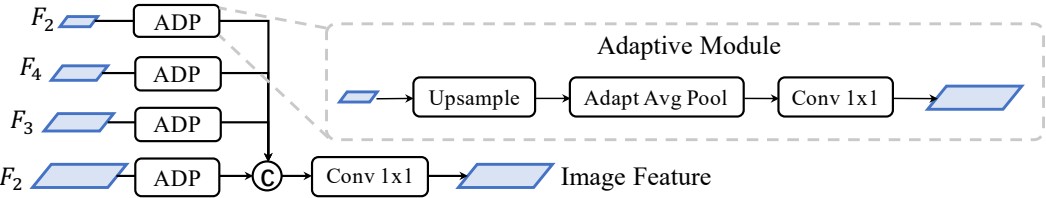

Figure 5: **Adaptive Module in FPN.**

## B   Experimental settings

**Evaluation metrics.** For 3D object detection, we report the official predefined metrics: mean Average Precision (mAP) [9], Average Translation Error (ATE), Average Scale Error (ASE), Average Orientation Error (AOE), Average Velocity Error (AVE), Average Attribute Error (AAE), and nuScenes Detection Score (NDS). The mAP is defined by the BEV center distance instead of the 3D IoU, and the final mAP is computed by averaging over distance thresholds of 0.5m, 1m, 2m, 4m across ten classes: car, truck, bus, trailer, construction vehicle, pedestrian, motorcycle, bicycle, barrier, and traffic cone. NDS is a consolidated metric of mAP and other indicators (e.g., translation, scale, orientation, velocity, and attribute) for comprehensively judging the detection capacity. The remaining metrics are designed for calculating the positive results' precision on the corresponding aspects (e.g., translation, scale, orientation, velocity, and attribute).

### B.1   Implementation details.

Our training consists of two stages: i) We first train the LiDAR stream and camera stream with multi-view image input and LiDAR point clouds input, respectively. Specifically, we train both streams following their LiDAR official settings in MMDetection3D [8]. Data augmentation strategies, e.g., random flips along the X and Y axes, global scaling and global rotation as data augmentation strategies are conducted for LiDAR streams. CBGS [67] is used for class-balanced sampling for CenterPoint and TransFusion. The camera stream inherit weights of backbone and neck from nuimage pre-trained Mask R-CNN CB-Swin-Tiny. ii) We then train BEVFusion that inherit weights from two trained streams. During fusion training, we set the learning rate to $1e^{-3}$ with a batch size of 32 on 8 V100 GPUs. For PointPillars, we train the detector for 12 epochs and reduce learning rate by $10\times$ at epoch 8 and 11. For CenterPoint and TransFusion, we train detectors for 6 epochs and reduce learning rate by $10\times$ at epoch 4 and 5, and we freeze the weights of the LiDAR stream for CenterPoint and TransFusion. Note that no data augmentation (i.e., flipping, rotation, or CBGS [67]) is applied when multi-view image input is involved. We only conduct BEV-space data augmentation when comparing with the state-of-the-art methods.

Table 8: **Ablating the Dynamic fusion module in LiDAR and camera malfunctions.**

| Dynamic Fusion | | Clean | | Object Failure | | Missing F | | Preserving F | | Stuck | |
| CSF | AFS | mAP↑ | NDS↑ | mAP↑ | NDS↑ | mAP↑ | NDS↑ | mAP↑ | NDS↑ | mAP↑ | NDS↑ |
|---|---|---|---|---|---|---|---|---|---|---|---|
| | | 64.9 | 69.9 | 34.6 | 53.6 | - | - | - | - | - | - |
| ✓ | | 67.3 | 70.5 | 50.1 | 57.5 | 65.4 | 70.5 | 63.5 | 68.7 | 65.6 | 69.9 |
| ✓ | ✓ | 67.9 | 71 | 50.3 | 57.6 | 65.9 | 70.7 | 65.1 | 69.9 | 66.2 | 70.3 |

Table 9: **Results when facing both camera and LiDAR malfunction.**

| Method | | clean | Object Failure + Missing F | Object Failure + Preserving F | Object Failure + Stuck |
|---|---|---|---|---|---|
| BEVFusion | mAP | 67.9 | 47.8 | 39.3 | 43.6 |
| | NDS | 71 | 56.2 | 52.9 | 54.8 |
| TransFusion | mAP | 66.9 | 34.2 | 33.6 | 33.9 |
| | NDS | 70.9 | 52.7 | 51 | 52.4 |

Table 10: **Results on robustness setting under different lighting conditions.**

| Method | Modality | Daytime | Nighttime |
|---|---|---|---|
| CenterPoint | L | 62.8 | 35.4 |
| TransFusion-L | L | 64.8 | 36.2 |
| TransFusion | LC | 67 | 41.8 |
| BEVFusion | LC | 68 | 42.4 |

When we finetune LiDAR-camera fusion detectors on the augmented training set, we train detectors for 12 epochs where the initial learning rate is set to $1e^{-4}$ and reduced by $10\times$ at epoch 8 and 11 with a batch size of 32.

# C  More robustness analysis

## C.1  Robustness of Dynamic Fusion Module

To better show the effectiveness of each part of the Dynamic Fusion Module, we test BEVFusion equipped with TransFusion-L and show the result under robustness settings against LiDAR and camera malfunctions in Sec. 4.4.1 and Sec. 4.4.2. We show the result in Table 8. We can see that when LiDAR fails to receive points from the object, with a simple channel& spatial fusion (CSF), BEVFusion greatly improves its LiDAR stream by 15.5% mAP. When adaptive feature selection (AFS) is adopted, the mAP can be further improved by 0.2%. Under camera missing scenarios, AFS improves CSF-only by 0.5-1.6% mAP. The results show that our Dynamic Fusion Module is still able to select the BEV information which exists to feed the final detection result under input malfunctions.

## C.2  Robustness under both modality malfunctions

We evaluate our BEVFusion equipped with TransFusion-L under the 'LiDAR fails to receive object reflection points' (as in Table 4), 'missing front camera', and 'preserving front camera' (as in Table 5) malfunctions and rereport the result in Table 9. The results show that BEVFusion remains effective in the face of a certain degree of two-modality malfunction. However, conceptually speaking, if one object is never captured by the camera or LiDAR, our framework will not be able to identify the object.

## C.3  Robustness against Inferior Image Conditions

We demonstrate the robustness of our proposed framework against inferior image conditions on nuScenesvalidation set. We report the results of BEVFusion equipped with TransFusion-L under different lighting conditions in Table 10. Compared with CenterPoint and TransFusion, BEVFusion shows the best robustness under different lighting conditions.

# D  Performance for different distance regions

We show the mAP results on different subsets based on the object distance range in Table 11 and Table 12. We compare BEVFusion equipped with CenterPoint, PointPillars, and TransFusion-L to its single modality

Table 11: **Results on the distance between object center and ego vehicle in meters.**

| Modality | | PointPillars | | | CenterPoint | | | TransFusion-L | | |
|---|---|---|---|---|---|---|---|---|---|---|
| Camera | LiDAR | <15m | 15-30m | >30m | <15m | 15-30m | >30m | <15m | 15-30m | >30m |
| | ✓ | 28.2 | 21.2 | 15.1 | 73.1 | 57.8 | 33.6 | 76.3 | 66.1 | 43.2 |
| ✓ | | 22 | 12.9 | 4.6 | 49.1 | 23.1 | 5.8 | 41.9 | 19.2 | 4.9 |
| ✓ | ✓ | 32.5 | 27.7 | 20.9 | 77.7 | 65 | 42.9 | 77.3 | 69.4 | 49.2 |

Table 12: **Distant regions performance comparison with TransFusion.**

| Method | overall | <15m | 15-30m | >30m |
|---|---|---|---|---|
| TransFusion | 65.6 | 75.5 | 66.9 | 43.7 |
| TransFusion (our implement) | 66.9 | 77.6 | 68.3 | 47.7 |
| BEVFusion | 67.9 | 77.3 | 69.4 | 49.2 |

Table 13: **Ablating the Dynamic fusion module in LiDAR and camera malfunctions.**

| Modality | | PointPillars | | CenterPoint | | TransFusion-L | |
|---|---|---|---|---|---|---|---|
| Camera | LiDAR | Mem. (MB) | Time (ms) | Mem. (MB) | Time (ms) | Mem. (MB) | Time (ms) |
| | ✓ | 7190 | 189.38 | 12468 | 199.87 | 12536 | 263.61 |
| ✓ | | 7948 | 1264.55 | 7956 | 1278.94 | 7950 | 1264.45 |
| ✓ | ✓ | 7968 | 1513.65 | 18078 | 1464.21 | 18086 | 1529.66 |

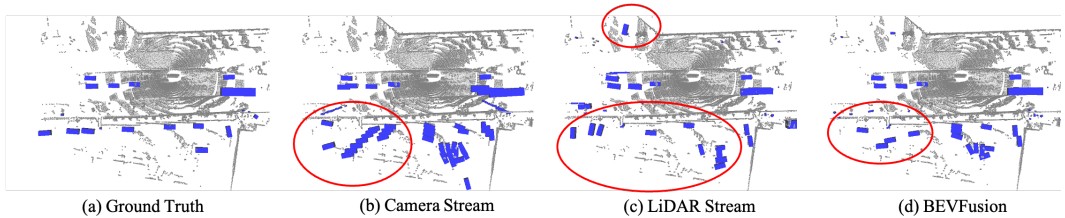

(a) Ground Truth  (b) Camera Stream  (c) LiDAR Stream  (d) BEVFusion

Figure 6: **Visualization of Failure cases of BEVFusion equipped with PointPillars as LiDAR stream.**

stream in Table 11. BEVFusion boosts its camera stream by 10%-35.4%, 14.8%-50.2%, and 16.3-44.3%mAP for distant regions in <15m, 15-30m, and >30m, respectively. BEVFusion boosts its LiDAR stream by 1%-4.6%, 3.3%-7.2%, 5.8%-9.3% mAP for distant regions in <15m, 15-30m, and >30m, respectively. We compare BEVFusion with TransFusion results reported from the original paper (12e + 6e training) and our re-implement results (20e + 6e training) in Table 12, where our BEVFusion surpasses TransFusion by 1.5% mAP for >30m distant regions. The results show that our fusion framework gives a larger performance boost for distant regions where 3D objects are difficult to detect or classify in LiDAR modality.

## E  Latency

We show the latency and memory usage of BEVFusion and its LiDAR and camera streams in Table 13. Latency is measured on the same machine with an Intel CPU (Xeon Gold 6126 @ 2.60GHz) and an Nvidia V100 GPU with a batch size of 1. Note that our latency bottleneck is the camera stream rather than our fusion framework (i.e., Dynamic Fusion Module). In our camera stream, the 2D->3D projector adopted from LSS [34] costs more than 957 ms, which can be improved through engineering deployment, i.e., concurrent processing.

## F  Failure cases analysis

We show the failure cases of BEVFusion equipped with PointPillars as the LiDAR stream in Fig. 6. Blue boxes are bounding boxes and the red-circled boxes are failed predictions. In Fig. 6.(b), the camera stream fails to predict the objects at the bottom-left of the BEV map, in (c), the LiDAR stream fails to predict the objects at the bottom and detects a false positive sample at the top-left, and in (d), BEVFusion fails to predict the objects at the bottom-left. These results imply that when one stream fails and the other succeeds in detecting objects, BEVFusion can balance the two streams well and succeed in detecting objects, but when both streams fail, BEVFusion also fails accordingly.