# OpenReview forum: "BEVFusion: A Simple and Robust LiDAR-Camera Fusion Framework"
_NeurIPS.cc/2022/Conference — NeurIPS 2022 Accept_

### Official Review · Reviewer_CaaL · 2022-07-04

**Rating:** 7
**Confidence:** 3
**Soundness:** 3 good
**Presentation:** 3 good
**Contribution:** 3 good

**Summary:**

Towards the problem of current methods tend to fail at situations where hardware malfunctions, this paper presents a simple yet effective LiDAR-Camera fusion framework, namely BEVFusion. By disentangling camera pipeline from LiDAR network and using a dynamic fusion module, BEVFusion achieves SOTA performance and shows robustness against LiDAR or camera malfunction at the same time. An effective modification on the camera pipeline is also proposed to boost the final performance.

**Questions:**

1. How does the baseline method in Table 7 fuse the features?
2. This does not affect my rating to the paper. Just out of curious, would BEVFusion still works when facing both camera and LiDAR malfunction?

**Limitations:**

The potential negative social impact is well discussed. But the limitation should be discussed more. For example, would the late-fusion style misses the opportunity to fuse intermediate LiDAR and camera features, and thus makes the pipeline suffer potential performance drop?

**Strengths And Weaknesses:**

## Strength
1. The paper is well written and easy to read.
2. Robustness of autonomous driving algorithms should be paid more attention to. This paper raises the issue and makes the attempt to addressing it.
3. Thorough experiments are performed. Claims are well-supported. SOTA performance is achieved on both normal and robust settings of nuScenes.
4. The clean design of the framework makes it easy to use any camera or LiDAR framework.


## Weaknesses
1. It is nice to see a simple yet effective module (dynamic fusion module) being proposed. But it would be nicer to provide some insights and analysis into the design itself. For example, by analyzing how would the fusion module work when facing incomplete LiDAR or camera inputs, we might gain some insights into the module design of CSF and AFS.
2. The experiments section does not provide runtime analysis, like inference time and memory footprint, and its comparison with other methods.

---

> ### Author Response · Authors · 2022-08-02
> **Response to Reviewer CaaL**
>
> We sincerely appreciate the positive feedback from the reviewer CaaL and provide detailed responses below.
> #### `Q1: ...(dynamic fusion module) ... provide some insights and analysis into the design itself... how would the fusion module work when facing incomplete LiDAR or camera inputs...`
> A1: Thanks for the suggestion. To better show the effectiveness of each part of the dynamic fusion module, we test BEVFusion equipped with TransFusion-L and show the result under robustness settings against LiDAR and camera malfunctions in Sec.4.4.1 and Sec.4.4.2. We show the result in the Table below.
>
> |     |     | \| | clean |      | \| | Object | Failure | \| | Missing  |   F  | \| | Preserving |   F  | \| | Stuck |      |
> |:---:|:---:|:--:|:-----:|:----:|:--:|:-------:|:-------:|:--:|:--------:|:----:|:--:|:----------:|:----:|:--:|:-----:|:----:|
> | CSF | AFS | \| |  mAP  |  NDS | \| |   mAP   |   NDS   | \| |    mAP   |  NDS | \| |     mAP    |  NDS | \| |  mAP  |  NDS |
> |  -  |  -  | \| |  64.9 | 69.9 | \| |   34.6  |   53.6  | \| |     -    |   -  | \| |      -     |   -  | \| |   -   |   -  |
> |  ✓  |  -  | \| |  67.3 | 70.5 | \| |   50.1  |   57.5  | \| |   65.4   | 70.5 | \| |    63.5    | 68.7 | \| |  65.6 | 69.9 |
> |  ✓  |  ✓  | \| |  67.9 | 71.0 | \| |   50.3  |   57.6  | \| |   65.9   | 70.7 | \| |    65.1    | 69.9 | \| |  66.3 | 70.2 |
>
>
> We can see that when LiDAR fails to receive points from the object, with a simple channel& spatial fusion (CSF), BEVFusion greatly improves its LiDAR  stream by 15.5% mAP. When adaptive feature selection (AFS) is adopted, the mAP can be further improved by 0.2%. Under camera missing scenarios, AFS improves CSF-only by 0.5-1.6% mAP. The results show that our dynamic fusion module is still able to select the BEV information which exists to feed the final detection results under input malfunctions.
> #### `Q2: The experiments section does not provide runtime analysis, like inference time and memory footprint, and its comparison with other methods. `
> A2: Thanks, we show the latency and memory usage of  BEVFusion and its LiDAR and camera streams in the Table below. Latency is measured on the same machine with an Intel CPU (Xeon Gold 6126 @ 2.60GHz) and an Nvidia V100 GPU with a batch size of 1.  Note that our latency bottleneck is the camera stream rather than our fusion framework (i.e., dynamic fusion module). In our camera stream, the 2D->3D projector adopted from LSS costs more than 957 ms, which can be improved through engineering deployment, i.e., concurrent processing.
>
> | Modality |  | &#124; | PointPillars |  | &#124; | CenterPoint  |  | &#124; | TransFusion-L  |  |
> | --- | --- | --- | --- | --- | --- | --- | --- | --- | --- | --- |
> | Camera | LiDAR | &#124; | Memory (MB) | Latency (ms) | &#124; | Memory (MB) | Latency (ms) | &#124; | Memory (MB) | Latency (ms) |
> |  | ✓ | &#124; | 7190 | 189.38 | &#124; | 12468 | 199.87 | &#124; | 12536 | 263.61 |
> | ✓ |  | &#124; | 7948 | 1264.55 | &#124; | 7956 | 1278.94 | &#124; | 7950 | 1264.45 |
> | ✓ | ✓ | &#124; | 7968 | 1513.65 | &#124; | 18078 | 1464.21 | &#124; | 18086 | 1529.66 |
>
>
> #### `Q3: How does the baseline method in Table 7 fuse the features?`
> A3: Thanks. In Table 7, the baseline is the LiDAR stream where only LiDAR is input, therefore no fusion is conducted. We will be more explicit in the revision.
> #### `Q4: ... would BEVFusion still works when facing both camera and LiDAR malfunction?`
> A4: Thanks. We evaluate our BEVFusion equipped with TransFusion-L under the 'LiDAR fails to receive object reflection points' (as in Table 4), 'missing front camera', and 'preserving front camera' (as in Table 5) malfunctions and rereport the result in the Table below. The results show that BEVFusion remains effective in the face of a certain degree of two-modality malfunction. However, conceptually speaking, if one object is never captured by the camera or LiDAR, our framework will not be able to identify the object.
>
> | Method      |     | clean | Object Failure | Missing F | Preserving F | Stuck | Object Failure + Missing F | Object Failure +  Preserving F | Object Failure +  Stuck |
> |-------------|-----|-------|----------------|-----------|--------------|-------|----------------------------|--------------------------------|-------------------------|
> | BEVFusion   | mAP | 67.9  | 50.3           | 65.9      | 65.1         | 66.2  | 47.8                       | 39.3                           | 43.6                    |
> |             | NDS | 71.0  | 57.6           | 70.7      | 69.9         | 70.3  | 56.2                       | 52.9                           | 54.8                    |
> | Transfusion | mAP | 66.9  | 34.6           | 65.3      | 64.4         | 65.9  | 34.2                       | 33.6                           | 33.9                    |
> |             | NDS | 70.9  | 53.6           | 70.1      | 69.3         | 70.2  | 52.7                       | 51.0                           | 52.4                    |

---

> ### Author Response · Authors · 2022-08-02
> **Response to Reviewer CaaL (Continued)**
>
> #### `Q5: More limitation discussions. The potential negative social impact is well discussed. But the limitation... For example, would the late-fusion style misses the opportunity to fuse intermediate LiDAR and camera features, and thus makes the pipeline suffer potential performance drop?`
> A5: Thanks for the insightful suggestions. Our fusion focuses on the high-level feature representations and it might miss the opportunity to fuse intermediate LiDAR and camera features. The alignment between intermediate LiDAR and camera features is an interesting topic and can be addressed in the future. We will add the discussion in the revision.

---

### Official Review · Reviewer_6d2t · 2022-07-10

**Rating:** 5
**Confidence:** 4
**Soundness:** 3 good
**Presentation:** 3 good
**Contribution:** 3 good

**Summary:**

Most existing camera-lidar fusion work decorates lidar points with image features and then performs detection in 3D/BEV space. This work leverages recent Lift-Splat-Shoot work for cameras, which allows one to map both camera and lidar inputs to BEV space, before fusing and applying the detection head.

**Questions:**

- Is the approach SOTA on nuscenes or not?  Can you please explicitly contrast your performance relative to the nuScenes leaderboard (at least for the published approaches).

- Can you provide an analysis of performance as a function of distance to object, and compare to a standard lidar approach and TransFusion/DeepFusion etc?

- Can you please provide results on at least one more dataset with high quality Lidar such as the Waymo Open Dataset?

- What is the latency of the approach and how does it compare to the baselines?

**Limitations:**

See comment on weaknesses. Some core potential limitations of the existing method have not been fully explored.

My current rating is predicated on the assumption that a similar idea has not been published yet (not completely certain) and that I will receive reasonable responses to my questions.

**Strengths And Weaknesses:**

Strengths:
- The proposed idea and its realization makes sense, and I am not aware of such published work (even though there seems to be concurrent similar work, since this seems a logical next step given the existence of LSS [52]).
- Details in the model seem well thought out. This include the extensions to LSS (Dual-Swin-Tiny architecture, ADP), as well as the layers in the dynamic fusion module.
- The experimental results show that this work is close to SOTA on nuScenes and that it affords significant model robustness in the case of lidar information missing compared to existing methods.
- The model details are pretty clearly explained.

Weaknesses:
- Related work section is confusing in a few places and can be streamlined further. Examples:
1) The Camera detectors section contains a discussion of PointPillars, which is a purely Lidar method.
2) Range images are not really Euclidean space (see line 88)
3) 89: "Recently, people start to exploit these two feature modalities to increase the representation power" --> There is earlier work to do this, if I understand correctly the statement. E.g. [5] from the paper, or End-to-End Multi-View Fusion for 3D Object Detection in LiDAR Point Clouds, by Yin Zhou et al, CoRL 2019.
4) 90: " Another line of work is to exploit the benefit of the bird’s eye view plane similar to the camera perception" --> a lot of this work came before camera started exploiting the BEV view.

- Intuitive explanations are lacking in a couple of instances:
1) Work does not explain the intuition why the model needs to be trained in two stages. What happens if it's trained in a single stage?
2) 14: "Note that we do not conduct data augmentation when multi-view image input is involved, while data augmentation plays a critical part in other cutting edge methods." Is this a limitation of camera fusion methods in general or something specifically lacking in your case? Can you please clarify?

- It is unclear whether the approach is SOTA on nuScenes or not. Can you please explicitly contrast your performance relative to the nuScenes leaderboard (at least for the published approaches). When exploring that leaderboard myself, I see mentions of a method called BEVFusion that is SOTA but seems to be a different method? Assuming that method is different and already on the leaderboard, your naming may be confusing / too generic.

- nuScenes is a dataset with particularly poor lidar (compared to other public datasets, such as Waymo Open Dataset, Argoverse2.0 etc). Results on at least one more dataset with high quality and longer-range lidar are highly desirable. The core issue of missing lidar points may be a lot less pertinent for more modern lidars. Also, as range increases beyond ~40m to 70-200m, the approach here may actually underperform lidar-painting approaches, since BEV view can start containing errors > 10m in the camera case making fusion in BEV space difficult. To this effect, analysis of the method performance as a function of object distance, relative to SOTA fusion methods for long distances will help.


Language:
There are minor language issues and typos in the paper, it would benefit from another proofreading pass.

---

> ### Author Response · Authors · 2022-08-02
> **Response to Reviewer 6d2t**
>
> We appreciate the detailed and constructive feedback from reviewer 6d2t. Please see our detailed responses below.
> #### `Q1: Related work section is confusing in a few places and can be streamlined further. The camera detectors section...Range images...There is earlier work to do this...a lot of this work came before camera started exploiting the BEV view.`
> A1: Thanks. We have fixed that in the revision.
> #### `Q2: Work does not explain the intuition why the model needs to be trained in two stages. What happens if it's trained in a single stage?`
> A2: Thanks for the suggestion. In the second stage of training, we can freeze the image-view encoder and 3D backbone to avoid accumulating gradients and save GPU memory and training time. For example, with the pre-trained TransFusion-L and TransFusion-camera, our BEVFusion needs only 8 hours of training on 8 V100 (32G) GPUs, while training from scratch requires at least 6 days on 8 A100 (80G) GPUs. Therefore, such two-stage training is easy to re-implement and memory-conserving.
> #### `Q3: 14: "Note that we do not conduct data augmentation when multi-view image input is involved..." Is this a limitation of camera fusion methods in general or something specifically lacking in your case? Can you please clarify?`
> A3: Adding camera augmentation requires further alignment between augmented RGB features and 3D ground truth boxes, and a joint LiDAR-camera augmentation is even more complicated. To maintain simplicity and fairly show the effectiveness and generalization ability of our framework, we do not rely on special training tricks. It is worth noticing that one setting of our method can achieve state-of-the-art without such augmentation.
> #### `Q4: It is unclear whether the approach is SOTA ... please explicitly contrast your performance relative to the nuScenes leaderboard (at least for the published approaches) ... your naming may be confusing / too generic.`
> A4: Thanks. Our result, 69.2% mAP, 71.8 NDS, achieves SOTA compared to published approaches and is publicly available on nuScenes leaderboard. We cannot reveal more information due to the double-blind policy. We kindly disagree with the comment about naming, it is a lovely coincidence that these two papers are concurrent works and release the results almost at the same time. Notably, our paper proposes a robust and general framework for LiDAR-camera fusion rather than a specific 3D perception method.
> #### `Q5: Can you provide an analysis of performance as a function of distance to object, and compare to a standard lidar approach and TransFusion/DeepFusion etc?`
> A5: Thanks for the suggestion. We show the mAP results on different subsets based on the object distance range in the two tables below. We compare BEVFusion equipped with CenterPoint, PointPillars, and TransFusion-L to its single modality stream below. BEVFusion boosts its camera stream by 10%-35.4%, 14.8%-50.2%, and 16.3-44.3%mAP for distant regions in <15m, 15-30m, and >30m, respectively. BEVFusion boosts its LiDAR stream by 1%-4.6%, 3.3%-7.2%, 5.8%-9.3% mAP for distant regions in <15m, 15-30m, and >30m, respectively.
>
> | Modality |  | &#124; |  | PointPillars |  | &#124; |  | CenterPoint |  | &#124; |  | TransFusion-L |  |
> | --- | --- | --- | --- | --- | --- | --- | --- | --- | --- | --- | --- | --- | --- |
> | Camera | LiDAR | &#124; | <15m | 15-30m | >30m | &#124; | <15m | 15-30m | >30m | &#124; | <15m | 15-30m | >30m |
> |  | ✓ | &#124; | 28.2 | 21.2 | 15.1 | &#124; | 73.1 | 57.8 | 33.6 | &#124; | 76.3 | 66.1 | 43.2 |
> | ✓ |  | &#124; | 22 | 12.9 | 4.6 | &#124; | 49.1 | 23.1 | 5.8 | &#124; | 41.9 | 19.2 | 4.9 |
> | ✓ | ✓ | &#124; | 32.5 | 27.7 | 20.9 | &#124; | 77.7 | 65 | 42.9 | &#124; | 77.3 | 69.4 | 49.2 |
>
> We compare BEVFusion with TransFusion results reported from the original paper (12e + 6e training) and our re-implement results (20e + 6e training) in the Table below, where our BEVFusion surpasses TransFusion by 1.5% mAP for >30m distant regions. The results show that our fusion framework gives a larger performance boost for distant regions where 3D objects are difficult to detect or classify in LiDAR modality.
>
> | Method | overall | <15m | 15-30m | >30m |
> | --- | --- | --- | --- | --- |
> | TransFusion | 65.6 | 75.5 | 66.9 | 43.7 |
> | TransFusion (our implement) | 66.9 | 77.6 | 68.3 | 47.7 |
> | BEVFusion | 67.9 | 77.3 | 69.4 | 49.2 |

---

> ### Author Response · Authors · 2022-08-02
> **Response to Reviewer 6d2t (Continued)**
>
> #### `Q6: Can you please provide results on at least one more dataset with high quality Lidar such as the Waymo Open Dataset?`
> A6: Thanks for your suggestion. We train BEVFusion equipped with PointPillars as LiDAR stream on WaymoD5-3classes and it barely improves the baseline. Due to the time constraints of rebuttal, we could not finetune the hyperparameters in detail and we will stress the problem in the future.
> As discussed in BEVFormer [1] and TransFusion [2], we suspect the reason is that  the camera system of Waymo can not capture the whole scene around the ego car, and thus the camera stream cannot perform full camera BEV space as it does on Nuscenes.  Furthermore, the camera-only detectors usually do not reach their potential on Waymo, i.e., BEVFormer achieves 0.069 L2/mAPH (IoU=0.7).
>
> [1] Zhiqi Li, et al. BEVFormer: Learning Bird's-Eye-View Representation from Multi-Camera Images via Spatiotemporal Transformers. In European Conference on Computer Vision (ECCV), 2022.
>
> [2] Xuyang Bai, et al. Transfusion: Robust lidar-camera fusion for 3d object detection with transformers.  In IEEE Conference on Computer Vision and Pattern Recognition (CVPR), 2022.
> #### `Q7: What is the latency of the approach and how does it compare to the baselines?`
> A7: Thanks, we show the latency and memory usage of  BEVFusion and its LiDAR and camera streams in the Table below. Latency is measured on the same machine with an Intel CPU (Xeon Gold 6126 @ 2.60GHz) and an Nvidia V100 GPU with a batch size of 1.  Note that our latency bottleneck is the camera stream rather than our fusion framework (i.e., dynamic fusion module). In our camera stream, the 2D->3D projector adopted from LSS costs more than 957 ms, which can be improved through engineering deployment, i.e., concurrent processing.
>
> | Modality |  | &#124; | PointPillars |  | &#124; | CenterPoint  |  | &#124; | TransFusion-L  |  |
> | --- | --- | --- | --- | --- | --- | --- | --- | --- | --- | --- |
> | Camera | LiDAR | &#124; | Memory (MB) | Latency (ms) | &#124; | Memory (MB) | Latency (ms) | &#124; | Memory (MB) | Latency (ms) |
> |  | ✓ | &#124; | 7190 | 189.38 | &#124; | 12468 | 199.87 | &#124; | 12536 | 263.61 |
> | ✓ |  | &#124; | 7948 | 1264.55 | &#124; | 7956 | 1278.94 | &#124; | 7950 | 1264.45 |
> | ✓ | ✓ | &#124; | 7968 | 1513.65 | &#124; | 18078 | 1464.21 | &#124; | 18086 | 1529.66 |

---

### Official Review · Reviewer_2Tdc · 2022-07-12

**Rating:** 5
**Confidence:** 4
**Soundness:** 3 good
**Presentation:** 3 good
**Contribution:** 2 fair

**Summary:**

The paper proposes a framework for 3D detection from RGB and LiDAR inputs in autonomous driving scenes. The pipeline includes separate networks reasoning from RGB and LiDAR inputs independently, and uses a fusion network for refined detection when both sources are available. Also the paper considers situations of data corruption and proposed to boost the robustness in the model design. The paper is the first to identify and evaluate the problem that most existing methods do not consider situations where one or both sources are unavailable, and proposes a pipeline customized for this situation. The proposed method is evaluated in the standard settings of object detection and compared with baseline methods both qualitatively and quantitatively.

**Questions:**

Please see the points in the Weakness section above.

**Strengths And Weaknesses:**

Strength:

[1] The task identification. As mentioned above, the paper is the first to identify the issue within the current literature and models, and proposes a pipeline accordingly which reasons from two sources independently and thus more robust when data unavailability occurs. In this sense, the task identification itself is valuable to the community in defining and bringing attention to the task.

[2] Extensive design choices and evaluation. Although the proposed pipeline is mostly based on existing methods, the paper is able to evaluate various design choices to demonstrate the flexibility of the proposed framework, as well as provide extensive evaluation into the results, yields SOTA results with both sources, and robust result when only one is available.

Weakness:

[1] Novelty and model design. The paper is novel in identifying the problem, which is legit and valuable. However for the proposed method itself, it is mostly a combination of existing methods utilization single sources without much modification, thus diminishing the merit of the proposed framework. Also the design to handle one or two sources in the framework is naive, basically running the first stage network only if only one source is available, and running both stages when two are available. A more sophisticated design could be, when for example camera stream is dropped for a few frames, is there a chance to stick to the fusion detector, but utilizing temporal information to compensate the missing RGB data, instead of simply drop the RGB branch and the fusion, running the LiDAR branch alone, which will likely result in a sudden drastic change to the detections?

[2] Simulation for data corruption. The paper proposes to augment the data to simulate possible data corruption scenarios, via dropping points and limiting FOV. However more effort can be done to boost the robustness: e.g. looking for real driving sequences in extreme weather or with bad data, and train/evaluate on those data.

---

> ### Author Response · Authors · 2022-08-02
> **Response to Reviewer 2Tdc**
>
> Thank you for your great efforts in the review of this paper. We are encouraged that the reviewer found our approach to be valuable to the community. We address the remaining concerns below.
>
> #### `Q1: Novelty and model design. The paper is novel in identifying the problem... However... it is mostly a combination of existing methods utilization single sources without much modification, thus diminishing the merit of the proposed framework. `
> A1: With respect, our framework is not a simple combination of existing methods.
> To improve the performance of the camera stream, we design the adaptive module in FPN, a simple BEV encoder, and CB-Swin-T as 2D backbone. To improve the performance of the fusion framework, we propose the dynamic fusion module to dynamically select feature fusion. Their improvements are shown in Tables 6 and 7. Furthermore, Our framework is generalized to multiple modern architectures, where the two streams can be replaced with various existing and future architectures. In the paper, we show the generalization ability of BEVFusion over different LiDAR streams, i.e., PointPillars, CenterPoint, and TransFusion.
> #### `Q2: The design to handle one or two sources in the framework is naive... A more sophisticated design could be, when for example camera stream is dropped for a few frames, is there a chance to stick to the fusion detector, but utilizing temporal information to compensate the missing RGB data, instead of simply drop the RGB branch and the fusion, running the LiDAR branch alone, which will likely result in a sudden drastic change to the detections?`
> A2: In our camera malfunctions experiments, the setting 'stuck' denotes that 50%  of camera frames are stuck. In such robust experiments, our framework inferences on two streams, where the camera input is the previous multi-view images from the latest available frame.
>
> #### `Q3: Simulation for data corruption...However more effort can be done to boost the robustness: e.g. looking for real driving sequences in extreme weather or with bad data...`
> A3: Thanks. We follow the suggestion and report the mAP of BEVFusion equipped with TransFusion-L under different lighting conditions in the Table below. Compared with CenterPoint and TransFusion, BEVFusion shows the best robustness under different lighting conditions.
> In the future, we will try to look for real driving sequences in extreme weather or with bad data, and train/evaluate those data.
>
> | Method |  Modality   | Daytime | Nighttime |
> | --- | --- | --- | --- |
> | CenterPoint | L | 62.8 | 35.4 |
> | TransFusion-L | L | 64.8 | 36.2 |
> | TransFusion | LC | 67.0 | 41.8 |
> | BEVFusion | LC | 68.0 | 42.4 |

---

### Official Review · Reviewer_fBas · 2022-07-12

**Rating:** 4
**Confidence:** 3
**Soundness:** 2 fair
**Presentation:** 2 fair
**Contribution:** 2 fair

**Summary:**

This paper introduced a method for point cloud object detection based on LiDAR camera fusion.  The main contribution of this method is the fusion framework that combines the camera and Lidar stream. This fusion module is very simple because it mainly consists of the concatenation of LiDAR and camera streams and a typical feature selection with an average pooling and 1x1 convolution. The results show that this method slightly outperformed the other method for the comparison. Moreover, the robustness against camera or LiDAR malfunctions is shown in the results. The ablation study shows that each module employed in this method improves performance.


**Questions:**

I am wondering why this simple fusion method is better than the other methods compare in this paper. If there are some results and discussions that make sense would be helpful for the readers.

**Ethics Review Area:**

["I don’t know"]

**Limitations:**

I would suggest showing some failure cases and discussions about them because it will contribute to the community.


**Strengths And Weaknesses:**

Strength
- The performance is slightly improved.
- The methodology is very simple.

Weakness
- Considering the small performance improvement and the simple methodology, I would think that the contribution of this method is relatively limited.

---

> ### Author Response · Authors · 2022-08-02
> **Response to Reviewer fBas**
>
> We thank reviewer fBas for the thoughtful comments. Please see our responses below.
> #### `Q1: Considering the small performance improvement and the simple methodology, I would think that the contribution of this method is relatively limited.`
> A1: To the best of our knowledge, this paper is the first to identify the downside within the current LiDAR-camera fusion models that inevitably fail when LiDAR input is missing and proposes a framework that disentangles the dependency of two sources, thus being more robust in the case of data unavailability. In this sense, task identification itself is valuable to the community in defining and bringing attention to the task. Furthermore, we propose extensive evaluation designs, yielding SOTA results with both clean and robust settings.  Despite the conceptual simplicity of the proposed framework, it is generative, effective, and robust. The simplicity does not diminish the contribution.
> #### `Q2: I am wondering why this simple fusion method is better than the other methods compare in this paper. If there are some results and discussions that make sense would be helpful for the readers.`
> A2: Thanks for your good question. Actually, our simple framework can surpass the state-of-the-art methods also surprises us, but this is exactly why our work is valuable to the research community. The essential reason for the superiority of our framework is not about simplicity, but about novelty. In fact, this simple approach has never been done in the current literature so no one knows this would work well until our paper.
> We suspect one reason that this simple approach is never tried before. When MVX-Net[1] first attempted to fuse camera and LiDAR information by using a LiDAR point to query the corresponding camera features, people try to improve the original framework by proposing superior components. In essence, almost all previous works can be categorized into that framework as shown in Figure 1 of the main paper.
> As we noticed that the previous frameworks have the downside of relying heavily on LiDAR input, we propose a completely different framework to fuse information at the BEV feature space. Moreover, we empirically show that a simple fusion module is adequate to surpass previous complex approaches, further evidence of the effectiveness of our proposed framework.
>
> [1] 	 Vishwanath A Sindagi, et al. MVX-Net: Multimodal voxelnet for 3d object detection. In International Conference on Robotics and Automation (ICRA), 2019.
> #### `Q3: I would suggest showing some failure cases and discussions about them because it will contribute to the community.`
> A3: Thanks. We show the failure cases of BEVFusion equipped with PointPillars as the LiDAR stream in Fig. 6 of the revised Appendix F. The blue boxes are bounding boxes and the red-circled boxes are failed predictions. In Fig.6.(b), the camera stream fails to predict the objects at the bottom-left of the BEV map, in (c), the LiDAR stream fails to predict the objects at the bottom and detects a false positive sample at the top-left, and in (d), BEVFusion fails to predict the objects at the bottom-left. These results imply that the proposed method can balance the two streams well and successfully detect objects when one stream fails and the other succeeds, but fails accordingly when both streams fail.

---

### Official Review · Reviewer_w9Mt · 2022-07-17

**Rating:** 6
**Confidence:** 3
**Soundness:** 3 good
**Presentation:** 3 good
**Contribution:** 3 good

**Summary:**

The authors propose a method to fuse two source of information for BEV detection namely multi-view images and LIDAR data in such a manner that any data defects in one source of information do not effect the network of the other method. Previous methods have combined the information from the two sources at different stages of the network pipeline, but they are prone to getting effected in the inference results when the data is corrupted from either source. This paper delays the combination of information to even later part of the pipeline thereby
mitigating the effect of bad/corrupted/unavailable data. They do so by generating a pseudo BEV point cloud just from multi-view cameras and combining that information with LIDAR BEV. The combination part is based on a dynamic fusion method which selects important fused features be it from camera based BEV or LIDAR based BEV. The results are shown where missing data in the LIDAR or camera image do not effect the detection unless both are missing for the same object in the scene e.g. car.

**Questions:**

For training the camera stream, a camera based BEV 2D point cloud is created. How is this camera stream trained as this may require a ground truth with camera based BEV 2D points and detections on that?


**Limitations:**

The reviewer didn't find any major limitations. The authors could discuss and show some failure cases of their work.

**Strengths And Weaknesses:**

Strength:
The paper addresses a problem which could be a real life problem in LIDAR or image based data capture, where the LIDAR data is missing due to 3D scene material/reflectance properties or the image could be missing from a video stream. These problems can cause existing networks to fail. This paper addresses this problem which can make the commercial deployment of such systems more doable. The paper is well written with clear explanation of the previous work. The results are detailed and show scenarios where they are better than previous best results in challenging situations.

Weakness:
1. The dynamic fusion module should be explained in more detail as its one of the contributions of this paper. It should be explained with a scenario where data is missing from either of the streams and how the formulation in Eq.1 and Eq.2 will still be able to select the BEV information which exists to feed the final detection result.
2. Citations 44-45, 32-33, 57-58 are repeated citations. Please fix it.
4. grammar: #92 start->started, #3 discover->discovered

---

> ### Author Response · Authors · 2022-08-02
> **Response to Reviewer w9Mt**
>
> We thank reviewer w9Mt for the insightful comments and time spent reviewing the paper.
> #### `Q1: The dynamic fusion module should be explained in more detail ... with a scenario where data is missing from either of the streams...will still be able to select the BEV information... `
> A1: Thanks for the suggestion! To better show the effectiveness of each part of the dynamic fusion module, we test BEVFusion equipped with TransFusion-L and show the result under robustness settings against LiDAR and camera malfunctions in Sec.4.4.1 and Sec.4.4.2. We show the result below, where TransFusion-L is the baseline.
>
> |     |     | \| | clean |      | \| | Object- | Failure | \| | Missing  |   F  | \| | Preserving |   F  | \| | Stuck |      |
> |:---:|:---:|:--:|:-----:|:----:|:--:|:-------:|:-------:|:--:|:--------:|:----:|:--:|:----------:|:----:|:--:|:-----:|:----:|
> | CSF | AFS | \| |  mAP  |  NDS | \| |   mAP   |   NDS   | \| |    mAP   |  NDS | \| |     mAP    |  NDS | \| |  mAP  |  NDS |
> |  -  |  -  | \| |  64.9 | 69.9 | \| |   34.6  |   53.6  | \| |     -    |   -  | \| |      -     |   -  | \| |   -   |   -  |
> |  ✓  |  -  | \| |  67.3 | 70.5 | \| |   50.1  |   57.5  | \| |   65.4   | 70.5 | \| |    63.5    | 68.7 | \| |  65.6 | 69.9 |
> |  ✓  |  ✓  | \| |  67.9 | 71.0 | \| |   50.3  |   57.6  | \| |   65.9   | 70.7 | \| |    65.1    | 69.9 | \| |  66.3 | 70.2 |
>
> We can see that when LiDAR fails to receive points from the object, with a simple channel& spatial fusion (CSF), BEVFusion greatly improves its LiDAR  stream by 15.5% mAP. When adaptive feature selection (AFS) is adopted, the mAP can be further improved by 0.2%. Under camera missing scenarios, AFS improves CSF-only by 0.5-1.6% mAP. The results show that our dynamic fusion module is still able to select the BEV information which exists to feed the final detection result under input malfunctions.
> #### `Q2: Citations 44-45, 32-33, 57-58 are repeated citations...grammar..`
> A2: Thanks, we have fixed that in the revision.
> #### `Q3: For training the camera stream...How is this camera stream trained as this may require a ground truth with camera based BEV 2D points and detections on that?`
> A3: Thanks. Training the camera stream relies on the same ground-truth annotation as the LiDAR stream, i.e. 3D bounding boxes, and the training does not require extra data. In the camera stream, the camera-view features are projected to the 3D ego-car coordinate features to generate pseudo voxels through the 2D->3D projector. In this way, the camera BEV feature is under the same feature space as the LiDAR BEV feature to share the same prediction head.
> #### `Q4: The authors could discuss and show some failure cases of their work.`
> A4: Thanks. We show the failure cases of BEVFusion equipped with PointPillars as the LiDAR stream in Fig. 6 of the revised Appendix F. The blue boxes are bounding boxes and the red-circled boxes are failed predictions. In Fig.6.(b), the camera stream fails to predict the objects at the bottom-left of the BEV map, in (c), the LiDAR stream fails to predict the objects at the bottom and detects a false positive sample at the top-left, and in (d), BEVFusion fails to predict the objects at the bottom-left. These results imply that the proposed method can balance the two streams well and successfully detect objects when one stream fails and the other succeeds, but fails accordingly when both streams fail.

---

### Author Response · Authors · 2022-08-02
**[Summary of Response] Thanks all reviewers for their thorough and insightful feedback!**

We thank all the reviewers for their time, insightful suggestions, and valuable comments. We are glad that all reviewers find our work is well motivated with valuable task definition (w9Mt, 2Tdc, 6d2t, CaaL),  simple and effective (w9Mt, fBas, 2Tdc, 6d2t, CaaL),with  clear explanation of the previous work (w9Mt, 2Tdc, CaaL), with comprehensive experiments on robustness (w9Mt, 2Tdc, 6d2t, CaaL), and detailed model explanation (6d2t, CaaL).

Before we respond to each reviewer's comments in detail, we revise the manuscript according to their suggestions, and we believe this makes our paper much stronger. Here is a list of changes we made:

1. In Abstract and Sec.2, we fix the typo and related work discussion.
2. In Appendix C.1, we add an ablation study of Dynamic Fusion Module under robustness settings.
3. In Appendix C.2 and C.3, we add experiments for robustness analysis on both modality malfunctions and inferior image conditions.
4. In Appendix D, we add experiments on the performance gain based on the object distance range.
5. In Appendix E, we provide latency and memory footprint comparisons.
6. In Appendix F, we provide more visualization results for failure cases and analysis.

Note that, in the revised version, we mark the text modification in blue.

---

### Meta-Review · Area_Chair_2gWi · 2022-08-23

**Recommendation:** Accept
**Confidence:** Certain

**Metareview:**

The paper proposes a method to fuse two sources of information for Bird’s Eye View (BEV) detection, namely multi-view images and LIDAR data, in a way that any data defects in one source of information does not affect the other. Most existing camera-lidar fusion works decorate lidar points with image features and then perform detection in 3D/BEV space. This work leverages recent Lift-Splat-Shoot work for cameras, which allows one to map both camera and lidar inputs to BEV space, before fusing and applying the detection head. The reviewers appreciate the identification of the problem of present fusion methods that are susceptible to damage in one of the two sources of information, the simplicity of the method and its good empirical performance. They raise concerns regarding its novelty, given  the obvious choices of the present method. The rebuttal submitted by the authors presents more empirical results and ablations. Most reviewers appreciate the contribution of the paper, and  the paper is suggested for publication.

**Award:**

No

---

### Decision · Program_Chairs · 2022-09-14

Accept